# Incorporation of Butanol into Nanopores of Syndiotactic Polystyrene

**DOI:** 10.3390/polym17222978

**Published:** 2025-11-08

**Authors:** Saki Fujino, Rei Miyauchi, Takahiko Nakaoki, Paola Rizzo

**Affiliations:** 1Department of Materials Chemistry, Ryukoku University, Seta, Otsu 520-2194, Japant23m059@mail.ryukoku.ac.jp (R.M.); 2Department of Chemistry and Biology and INSTM Research Unit, University of Salerno, Via Giovanni Paolo II 132, 84084 Fisciano, Italy; prizzo@unisa.it

**Keywords:** syndiotactic polystyrene, nanopores, infrared spectroscopy, butanol, diffusion coefficient

## Abstract

Biobutanol can be obtained by fermentation of microorganisms and used as biofuel. The membrane separation is energetically favorable. The incorporation of butanol into syndiotactic polystyrene (sPS) with crystalline nanopores was investigated as a function of the butanol uptake temperature using infrared spectroscopy. The OH stretching modes at 3596 and 3300 cm^−1^, corresponding to hydrogen-bonded butanol in the crystalline cavity and free butanol in the amorphous region, respectively, were employed for analysis. Upon immersion of the sPS film in butanol, butanol molecules were absorbed in the crystalline nanocavities and amorphous phase. Diffusion increased with the uptake temperature in both regions. This can be associated with the larger molecular mobility of butanol molecules at high temperatures, facilitating contact between the film surface and the butanol molecules. The number of butanol molecules incorporated into the crystalline cavity was estimated using Lambert-Beer’s law. On average 90% of the nanopore cavities were occupied by butanol, while the remaining 10% were empty.

## 1. Introduction

Recent energy-related concerns have sparked interest in bio-based materials from both environmental and economic perspectives. Biofuels, such as bioethanol and biobutanol, fermented from natural products by microorganisms, are key materials for addressing these issues. Currently, bioethanol is the most commonly used biofuel. Its biosynthetic process is widely known, and it can be obtained in concentrations as high as 10–20% throughout the brewing process. Biobutanol has a higher molar enthalpy than ethanol because it comprises four carbons (compared with two carbons in ethanol), as well as a higher boiling temperature than that of ethanol, corresponding to lower volatility [1,2,3,4,5,6]. However, the biggest problem is that butanol concentrations are as low as 1–2% when fermented by microorganisms. This is because butanol is more toxic to microorganisms and restricts their growth. These alcohols can be separated from water by distillation; however, this approach has an energy disadvantage owing to the high heat capacity of water. Therefore, film separation must be considered. This not only reduces carbon dioxide emissions but also oil consumption. Nanoporous syndiotactic polystyrene (sPS) is a promising candidate for separating alcohol from water. The formation of clusters between butanol and water via hydrogen bonding may play an important role in the adsorption of butanol into nanoporous sPS. However, this report focuses on pure butanol as a preliminary investigation. Additionally, byproducts such as ethanol and acetone produced by microorganisms can also be incorporated into nanoporous crystalline phase of sPS and subsequently separated by distillation.

After the successful preparation of sPS in 1986 [7,8], several studies have analyzed its molecular structure [9,10,11,12,13,14,15,16,17,18,19,20,21,22,23,24,25,26,27,28,29,30,31,32,33,34,35,36,37,38,39,40,41,42,43,44,45,46,47,48,49,50,51,52,53]. In particular, two crystalline forms, α [9,10,11,12,13,14,15,16,17] and β [18,19], presenting polymer chains in all-*trans* conformations, and three crystalline forms—γ [20], δ [20,21,22,23,24,25,26,27,28,29,30,31,32], and ε [33]—with chains in s(2/1)2 helical conformation are described in [9,10,11,12,13,14,15,16,17,18,19,20,21,22,23,24,25,26,27,28,29,30,31,32,33]. Notably, the δ- and ε-forms can co-crystallize with low-molecular-weight compounds [34,35,36,37,38,39,40,41,42,43,44,45,46,47,48,49,50,51,52,53,54,55,56,57,58] such as hexanal [42], carboxylic acid as hexanoic acid [43], benzoic acid [44], salicylic acid [45], adipic and stearic acid [46], phenols as carvacrol [47], and thymol [48] owing to the large gaps between helices in the crystal units. After removal of low-molecular-weight molecules by suitable procedures (for instance, using treatment with volatile solvent as acetonitrile or supercritical CO_2_), the nanoporous crystalline (NC) phases (δ and ε) remain essentially unaltered, in terms of degree of crystallinity compared to the pristine cocrystalline phase, and can be easily reused [20,21,22,23,24,25,26,27,28,29,30,31,32]. We reported the incorporation of alcohols such as methanol and pentanol into the crystalline cavity of δ-sPS [59]. Alcohols with few carbons were quickly included into the nanopores of sPS, whereas branched alcohols, such as *tert*-butanol, could not be incorporated owing to their bulkiness. The former can occupy not only the crystalline cavity but also the amorphous region. Guerra et al., adsorbing monoterpenoid phenols such as carvacrol [47] into nanoporous sPS from dilute carvacrol/acetone solutions, showed that carvacrol molecules in the amorphous region progressively moved into the crystalline nanocavities when the solvent carrier (acetone) was desorbed. The carvacrol content, after equilibrium sorption from 5 wt % carvacrol solution in acetone in both the crystalline and amorphous regions was approximately 40 wt.

In this study, the adsorption of butanol into the nanopores of δ-sPS was investigated as a function of temperature to accelerate the incorporation.

## 2. Materials and Methods

sPS was supplied by Idemitsu Petrochemical, Tokyo, Japan. The molecular weight average (*M*_w_) and polydispersity index (*M*_w_/*M*_n_) were 2.4 × 10^5^ and 2.3, respectively. A δ-sPS film was prepared by treating the film obtained from 5% chloroform solution in supercritical carbon dioxide (scCO_2_). scCO_2_ was treated using a supercritical fluid apparatus (Jasco SCF-Get, Tokyo, Japan) under mild conditions of 40 °C and 10 MPa. The film, with a thickness of 40 μm, was cut into 1.5 cm squares. The NC δ sPS film was then immersed in 100 mL of butanol at the temperatures of 30, 40, 50, and 60 °C, then FT-IR measurements were performed. The measurements were performed until butanol sorption reached a plateau.

Wide-angle X-ray diffraction (XRD) patterns were obtained using an automatic Rigaku RINT2500, Tokyo, Japan diffractometer with Ni-filtered CuKα radiation. The 2θ scanning range was 2–35°, with a step of 0.02° and a scan speed of 2°/min.

Infrared (IR) spectra were measured using a JASCO FT-IR660 Plus, Tokyo, Japan spectrometer in the standard wavenumber range of 400–4000 cm^−1^. Transmission measurements were performed at room temperature, and 32 transients were collected for each spectrum, at a resolution of 1 cm^−1^. The thickness of the cast film was calibrated using the CH stretching mode of the phenyl ring at 3082 cm^−1^ for sPS as an internal standard [60].(1)$$A_{3082}=1.004\times 10^{-2}l$$
where *A*_3082_ and *l* (μm) denote the absorbance at 3082 cm^−1^ and film thickness, respectively. The reduced absorbance of the vibrational mode was obtained by normalizing to 50 μm thickness.

## 3. Results and Discussion

### 3.1. XRD Patterns of the sPS Film Before and After Immersion in Butanol

The XRD patterns of the sPS films before and after immersion in butanol are provided in Figure 1. As shown in Figure 1(a), the typical XRD pattern of the NC δ phase is characterized by (010) crystallographic planes at 2ϑ = 8.4°. After immersion of the NC δ film in butanol at 60 °C for 1 week, the peak at 2ϑ = 8.4° shifted to 2ϑ = 8.1°, Figure 1(b), indicating the incorporation of butanol molecules into the sPS nanoporous crystalline phase, leading to the formation of co-crystallized (CC) phases.

### 3.2. Diffusion of Butanol into the sPS Film

Figure 2 shows the IR spectra of the NC δ films before and after immersion in butanol for 5 h at varying temperatures, in the range of 3000–4000 cm^−1^.

A broad band at 3300 cm^−1^, characteristic of the OH stretching mode of butanol, as well as a narrow peak at 3596 cm^−1^, arise when butanol molecules are absorbed in the NC δ films. These two IR peaks, associated with the OH stretching modes of butanol molecules, are comparable to those reported in a previous paper on the incorporation of ethanol in nanoporous sPS samples [59], as well as in recent papers on the inclusion of carboxylic acids (e.g., hexanoic acid [42], benzoic acid [43], salicylic acid [44]) in the crystalline cavities of sPS samples. In particular, when butanol molecules are incorporated into δ crystalline nanocavities, the molecules are isolated from each other and do not interact via hydrogen bonding, giving rise to the narrow OH stretching peak at 3596 cm^−1^. Meanwhile, the space in the amorphous region, corresponding to the free volume, is sufficiently large for butanol molecules to form hydrogen bonds with each other, yielding a broad band at 3300 cm^−1^, as observed in the IR spectrum of liquid butanol. In the case of the OH stretching modes of ethanol molecules adsorbed in NC δ sPS, peaks were observed at 3360 and 3650 cm^−1^, which are at higher wavenumbers than those of butanol molecules. This difference can be attributed to variations in the vibrational strength of the OH mode between ethanol and butanol, as well as differences in their hydrogen bonding states. With increasing uptake temperature of butanol, the intensities of the peaks at 3300 and 3596 cm^−1^ increased, indicating that a larger amount of butanol was incorporated into the crystalline cavities and amorphous regions, and that the soaking temperature is important in facilitating butanol incorporation. The absorbance of butanol molecules adsorbed onto the NC δ films at varying temperatures is plotted as a function of time in Figure 3.

For the 3596 cm^−1^ band at 30 °C, more than 40 h were required to reach the equilibrium state, whereas less than one hour was required at 60 °C. Similarly, the adsorption of butanol into the amorphous region, represented by the band at 3300 cm^−1^, was faster at elevated temperature. In Figure 4, the plot of reduced absorbance (*A*(*t*)/*A*_0_) against the square root of time (t^1/2^) was evaluated for the butanol peaks at (A) 3596 cm^−1^ and (B) 3300 cm^−1^ at varying temperatures.

The diffusion coefficient, D, was estimated using the Fickian equation:(2)$$\frac{A(t)}{A_{0}}=\frac{4}{d}\sqrt{\frac{Dt}{\pi }}$$
where *A*(*t*) and *A*_0_ denote the measured absorbance as a function of time and at equilibrium, respectively; and *d* denotes the film thickness. Butanol diffuses towards the back of the film. The diffusion coefficients obtained for butanol molecules absorbed into the NC δ and amorphous phases of sPS films at 30 °C are 0.366 × 10^−10^ cm^2^ s^−1^ and 0.155 × 10^−10^ cm^2^ s^−1^, respectively. Kinetic sorption of butanol molecules was faster in the crystalline cavity than in the amorphous region. In Figure 5, the diffusion coefficients for butanol molecules in the NC δ and amorphous phases of the sPS films are plotted against the soaking temperature.

The slope became steeper with increasing temperature in both regions, indicating that higher temperatures provided larger diffusion coefficients. In particular, the diffusion coefficient of butanol molecules in the NC δ phase, which was obtained from the peak at 3596 cm^−1^, increased by ~30-fold, from 0.366 × 10^−10^ cm^2^ s^−1^ at 30 °C to 10.5 × 10^−10^ cm^2^ s^−1^ at 60 °C. Similarly, the diffusion coefficient of butanol molecules in the amorphous phase increased from 0.155 × 10^−10^ cm^2^ s^−1^ to 30 °C to 7.2 × 10^−10^ cm^2^ s^−1^ at 60 °C, corresponding to an ~45-fold increase. However, the diffusivity of butanol into NC δ phase is always higher than that of butanol molecules into amorphous sPS phase.

### 3.3. Weight Fraction of Butanol Molecules Incorporated into the Crystalline Cavity of sPS

Figure 6 plots the absorbance of the butanol peak at 3596 and 3300 cm^−1^ at equilibrium after enough time has passed versus the soaking temperature. The equilibrium absorbance at 3596 and 3300 cm^−1^ corresponds to the maximum amount of butanol in the crystalline and amorphous regions, respectively.

The equilibrium absorbance was almost the same, irrespective of the soaking temperature. According to the Beer–Lambert law, the absorbance is proportional to the density (*c*), as follows:(3)A=εcl,
where *A*, ε, and *l* denote the absorbance, absorption coefficient, and film thickness, respectively. Therefore, the equilibrium absorbance, irrespective of temperature, corresponds to the same amount of butanol incorporated into the crystalline cavity, although the diffusion coefficient depends on the temperature. The density of butanol incorporated into sPS can be estimated using Equation (3); however, the absorption coefficient for the OH stretching mode without hydrogen bonding (3596 cm^−1^ peak) is required. This absorption coefficient was estimated as follows. Butanol was mixed with a large amount of toluene (1 wt% butanol in toluene) to reduce the contact among butanol molecules. Figure 7 shows the OH peaks of butanol solution in toluene (1 wt%), butanol, and butanol incorporated into sPS.

Notably, only one OH peak (3596 cm^−1^) was observed; the peak position was exactly the same as that of butanol absorbed into NC δ sPS films. The absorbance of the OH stretching mode of 1 wt% butanol/toluene solution was plotted as a function of the spacer thickness (Figure 8).

The slope corresponds to the absorption coefficient, estimated to be 1.82 × 10^−3^ cm^2^ mol^−1^. As shown in Figure 6, the equilibrium absorbance for 3596 cm^−1^ was ~0.51. From Equation (3), the density of butanol in the film was estimated to be 8.8 × 10^−2^ g/cm^3^. As reported in our previous study [61], the density of the film without solvent molecules was 1.04 g/cm^3^. Therefore, 7.8 wt% butanol in 1 g sPS was incorporated into the nanocrystalline pores of the δ-sPS film.

### 3.4. Number of Butanol Molecules in the Crystalline Cavity of δ-sPS

The number of butanol molecules in the crystalline cavity was estimated from the density of butanol in the NC δ phase. The butanol molecules that absorbed at 3596 cm^−1^ would be incorporated into the crystal nanopores. Two eight-shaped chains and nanopores were observed in the unit cell. The number of butanol molecules in a crystal unit (n_but_) is expressed as follows [60]:(4)$$n_{but}=\frac{8c_{but}}{c_{sPS}\cdot x_{c}},$$
where *c_sPS_* and *c_but_* denote the molar concentrations of the sPS film and butanol in the film, respectively, as described in the literature [61]; and *x_C_* denotes the degree of crystallinity, which was determined to be 38% based on the peak at 572 cm^−1^ characteristic of δ-sPS in the IR spectrum [61]. Using this *x_C_* value, the number of butanol molecules was estimated to be 1.74 in the crystal unit, corresponding to ~0.9 butanol molecules incorporated into one nanopore of the crystal. Our previous study on the incorporation of ethanol showed that 1.9 ethanol molecules are incorporated into one pore [60]—almost twice that obtained for butanol (~0.9)—and can be explained by the smaller molecular size of ethanol than that of butanol. The value of 0.9 is close to 1, indicating that most nanopores were occupied by butanol molecules and that not all crystalline cavities were filled by butanol molecules. Namely, ~90% of the nanopores were occupied by butanol molecules, and the remaining 10% were empty.

## 4. Conclusions

The incorporation of butanol into the crystalline cavities of sPS films was investigated using FTIR spectroscopy. When the sPS films were immersed in butanol, IR peaks at 3596 and 3300 cm^−1^—associated with the OH stretching modes of butanol with and without hydrogen bonding, respectively—are apparent, corresponding to the incorporation of butanol molecules in both the crystalline cavity and amorphous region of the sPS films, respectively.

With increasing temperature of immersion in butanol, butanol incorporation was faster and almost complete within a few hours. The diffusion coefficient of butanol in the crystalline nanocavity was always higher than that in the amorphous phase at all temperatures investigated. The equilibrium absorbance was almost identical, irrespective of the uptake temperature, indicating that the same amount of butanol was incorporated because the crystallinity of sPS was the same. The average number of butanol molecules per nanopore was 0.9 compared to 1.9 for ethanol molecules. This difference is attributed to the smaller molecular size of ethanol compared to butanol.

## Figures and Tables

**Figure 1 polymers-17-02978-f001:**
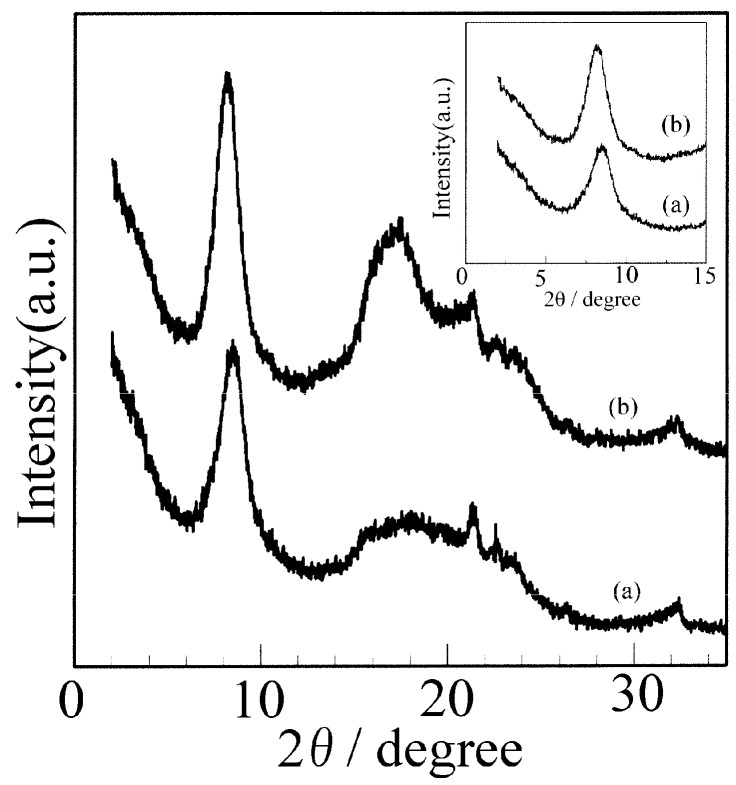
XRD patterns of the sPS film before and after immersion in butanol: (a) NC δ phase and (b) CC phase.

**Figure 2 polymers-17-02978-f002:**
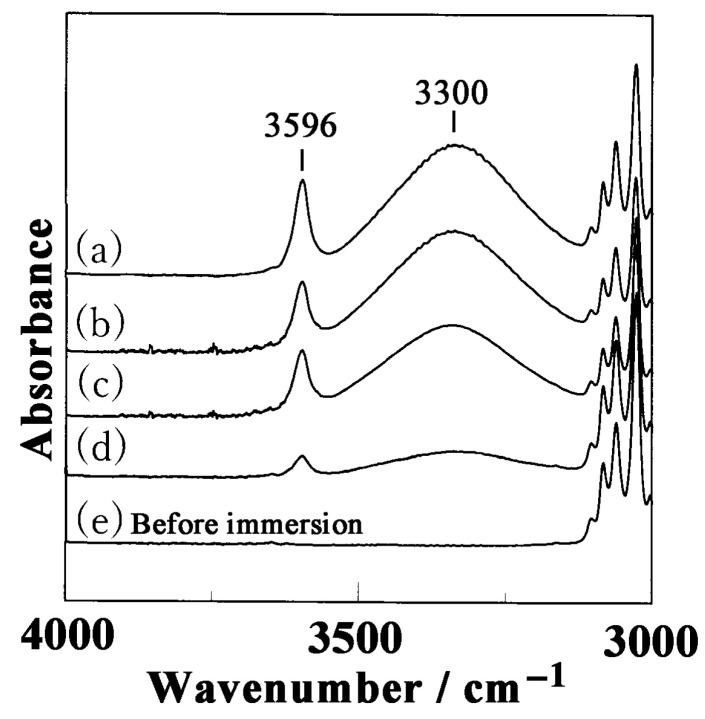
OH stretching modes of butanol in IR spectra of NC δ sPS films immersed in butanol for 5 h: (a) 60 °C, (b) 50 °C, (c) 40 °C, and (d) 30 °C. (e) IR spectrum of NC δ film before immersion in butanol.

**Figure 3 polymers-17-02978-f003:**
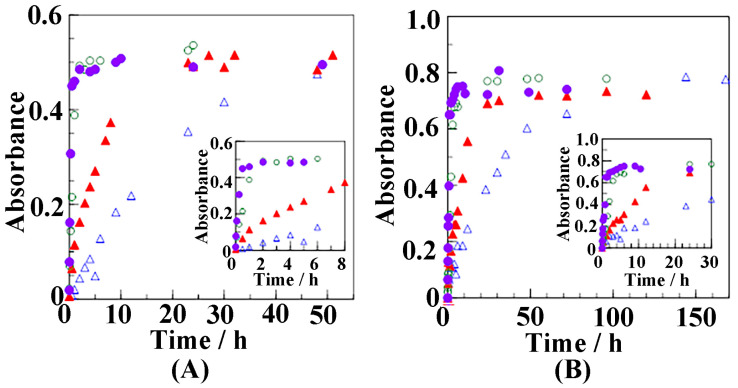
Absorbance peaks at (**A**) 3596 cm^−1^ and (**B**) 3300 cm^−1^ of butanol molecules adsorbed onto NC δ sPS films as a function of time at varying temperatures, where ∆: 30 °C, ▲: 40 °C, ○: 50 °C, and ●: 60 °C.

**Figure 4 polymers-17-02978-f004:**
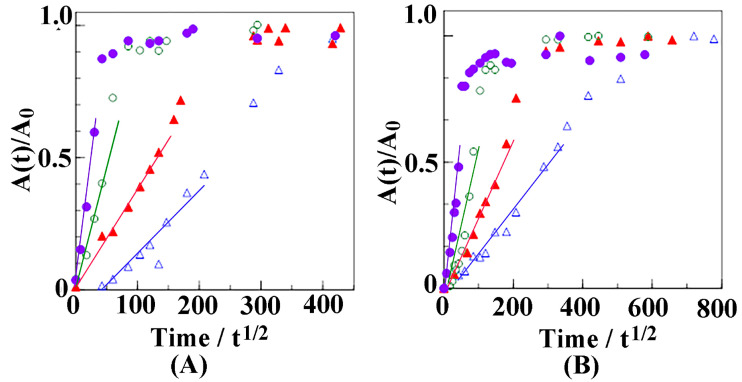
Plots of *A(t)*/*A*_0_ vs. t^1/2^ at (**A**) 3596 cm^−1^ and (**B**) 3300 cm^−1^ as a function of time for the films immersed in butanol. ∆: 30 °C, ▲: 40 °C, ○: 50 °C, and ●: 60 °C.

**Figure 5 polymers-17-02978-f005:**
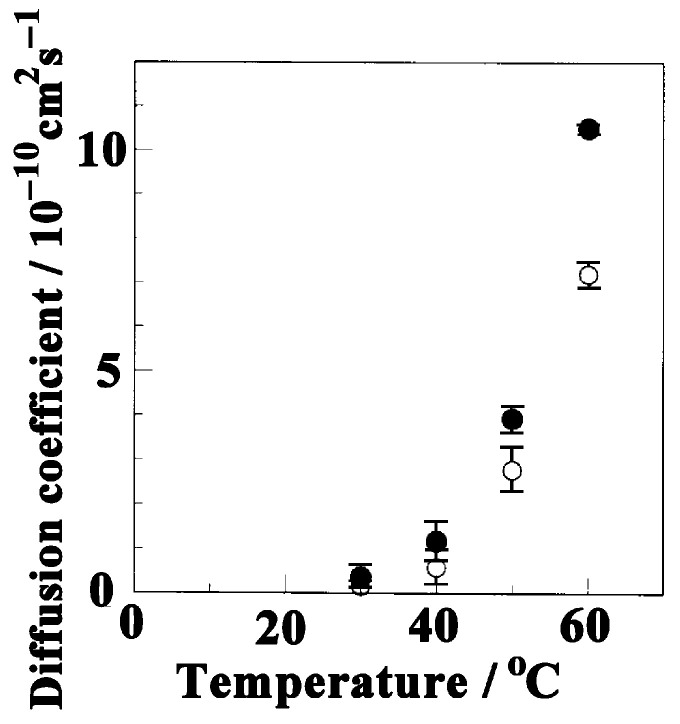
Diffusion coefficients of butanol molecules in the NC δ (●) and amorphous (○) phases of sPS films as a function of temperature.

**Figure 6 polymers-17-02978-f006:**
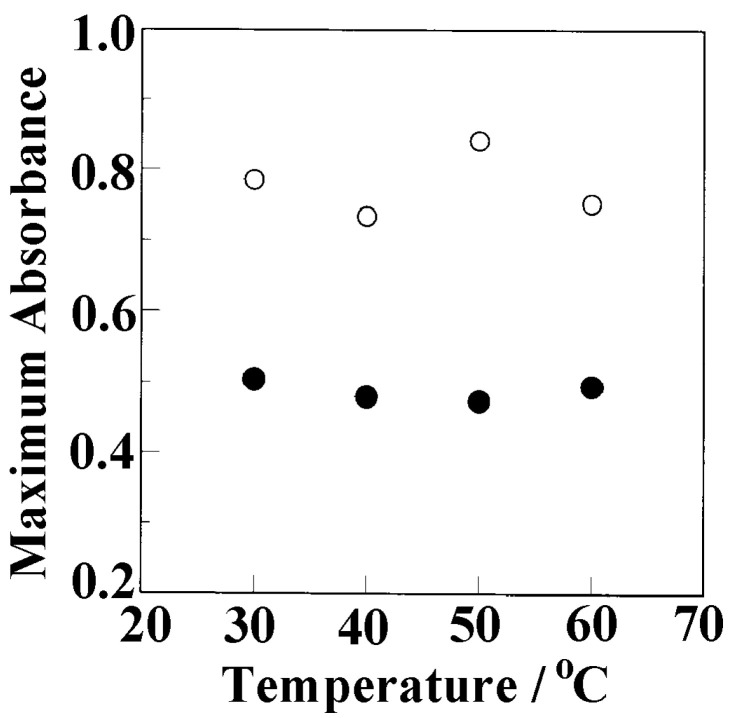
Absorbance of butanol peaks at 3596 cm^−1^ (●) and 3300 cm^−1^ (○) at equilibrium as a function of soaking temperature.

**Figure 7 polymers-17-02978-f007:**
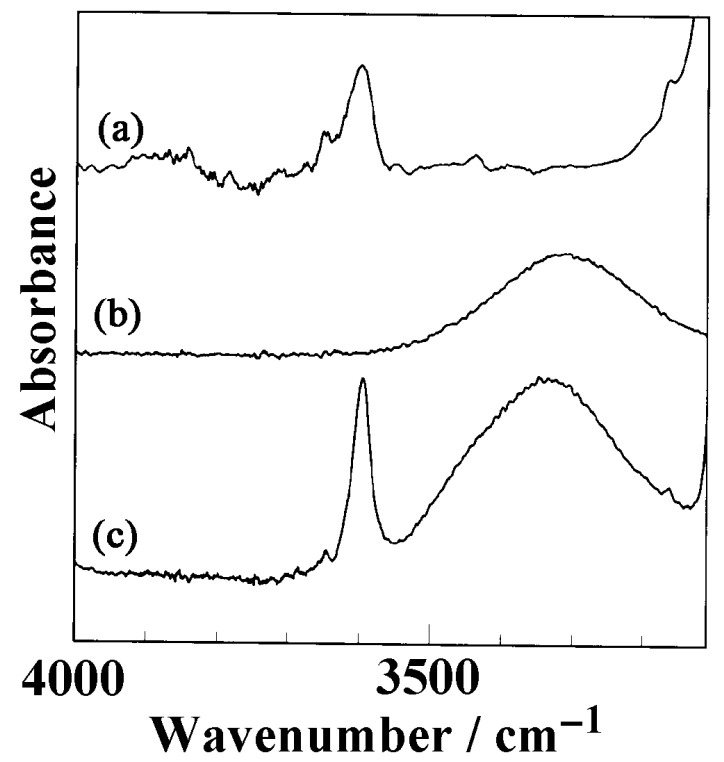
OH stretching region of IR spectra of (a) 1 wt% butanol in toluene, (b) liquid butanol, and (c) butanol absorbed into NC sPS films.

**Figure 8 polymers-17-02978-f008:**
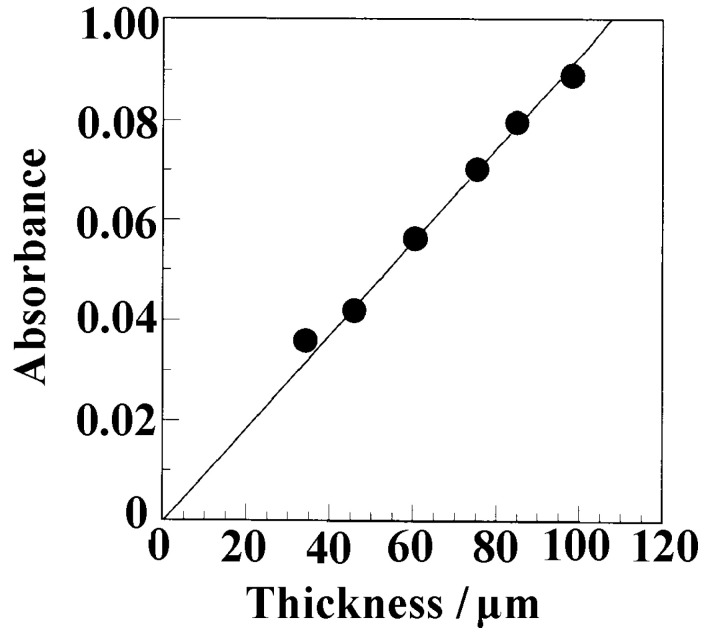
Absorbance of OH stretching mode of 1 wt% butanol in toluene (3596 cm^−1^) as a function of spacer thickness.

## Data Availability

Data are contained within the article.

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
