# Peer review of "Incorporation of Butanol into Nanopores of Syndiotactic Polystyrene"

_polymers, 2025, doi:10.3390/polym17222978_

Round 1

Reviewer 1 Report

Comments and Suggestions for Authors

The incorporation of butanol into syndiotactic polystyrene (sPS) with crystalline nanopores was investigated as a function of butanol uptake temperature by infrared spectroscopy. The OH stretching modes of butanol at 3,596 and 3,300 cm−1 bands with and without hydrogen bonds in the crystalline cavity and amorphous region, respectively, were employed for analysis. When the sPS film was immersed in butanol, butanol molecules were incorporated into the crystalline cavity and amorphous region. The diffusion increased with uptake temperature for both regions. The number of butanol molecules incorporated in the crystalline cavity was estimated by Lambert–Beer’s law. On average, 87% of the cavity nanopores were occupied by butanol and the remaining 13% were empty. The idea suggested in the manuscript is useful. However, the characterization is not enough in the manuscript. The morphology of the nanopore structure of sPS is suggested to provided. The diffusion process as well as diffusion coefficient of butanol are suggested to use more precise method, e.g., pulsed field gradient nuclear magnetic resonance. Major revision is necessary before it can be accepted.

Reviewer 2 Report

Comments and Suggestions for Authors

The manuscript reports on the incorporation of butanol into syndiotactic polystyrene with crystalline cavities and amorphous regions, as a function of temperature.

The work is interesting with enough data to support the conclusions. 

Questions: 

  1. The intended application of this material is to recover alcohols, in this case biobutanol from aqueous reaction mixtures obtained from fermentation. But this study focused on adsorption of pure butanol (not aqueous solutions). How do you expect this material to behave in 1-2% butanol aqueous solutions? Will water compete for adsorption? what about other byproducts from fermentation? A paragraph with appropriate references discussing this point should be added, either in the introduction or in the discussion.
  2. The format of the figures, in particular the graphs should be homogenized, font size is different in all of them, axis graduation marks, use of bold font, etc. 
  3. No details have been given about how the experiments are conducted, it is only mentioned: The NC δ sPS film was immersed in butanol at desired temperature. No details on the size, shape, weight and thickness of the films and the amount of butanol in which these are immersed. This should be added in the Materials and Methods section. 

Reviewer 3 Report

Comments and Suggestions for Authors

Review of the manuscript “Incorporation of Butanol into Nanopores of Syndiotactic Polystyrene”

I read the manuscript with interest. The authors present a solid piece of work on butanol sorption into δ-syndiotactic polystyrene. The combination of XRD and IR measurements is convincing, and the results are clearly described. Overall the paper is good and worth publishing, but there are a few places where some clarifications or small improvements would help the reader.

Comments:

  1. In the abstract (p.1, lines 10–23) it would be useful to say a word about why this study matters beyond the lab, e.g. for alcohol–water separation or fuel applications.

  2. The introduction (p.2, lines 35–40) is fine, but could mention more directly what’s missing in current separation methods and how sPS actually helps.

  3. In methods (p.3, lines 68–74) the immersion time at each temperature is not completely clear. Please give those numbers.

  4. Figure 2 (p.4) would be easier to follow if the OH peaks at 3596 and 3300 cm⁻¹ were marked, and the “before immersion” spectrum was clearly labeled.

  5. When discussing IR data (pp.4–5), it might be helpful to point back to the ethanol case (ref. 35) since the comparison is natural here.

  6. For Equation (2) (p.5), could the authors state which geometry assumption was used? Readers not familiar with diffusion models might get lost.

  7. In Figure 5 (p.6) it would help if some indication of error or uncertainty was given for the diffusion coefficients.

  8. On p.8 (lines 195–205), where the number of incorporated molecules is calculated, it would be clearer if the 38% crystallinity was written directly in the text instead of only in a reference.

  9. The conclusions (pp.9, 220–233) could stress more directly the practical difference between ethanol and butanol, the size effect on pore occupancy is interesting and deserves a sentence or two.

  10. The funding note (p.10, 238–243) could acknowledge more clearly the international cooperation, since that is actually one of the strengths of the study.

Reviewer 4 Report

Comments and Suggestions for Authors

The authors presented an interesting study on the incorporation of butanol into syndiotactic polypropylene. The work is undoubtedly relevant, but seems insufficiently substantiated:

  1. In what mode were the IR spectra obtained? Transmission? Or ATR?
  2. The black-and-white figures presented by the authors are quite difficult to understand.
  3. Fig. 3 – why is there a clear trend for the curves to change when moving from 30 to 40°C and then to 50°C, while the data for 50-60°C are identical, and in some places the absorbance at 50°C are higher than it at 60°C? Moreover, this effect is clearly visible not only for the 3596 cm-1 band, but also for the 3300 cm-1 band. The authors should provide an explanation in the text.
  4. The authors should explain why temperatures of 30, 40, 50, and 60°C were chosen for the measurements.
  5. What is the measurement error for diffusion coefficients using this method? Can we say that diffusion coefficients of 0.366 × 10−10 cm2 s−1 and 0.155 × 10−10 cm2 s−1 really differ enough to indicate that butanol sorption in the crystalline and amorphous phases differs? Have the obtained diffusion coefficients been compared using other methods?
  6. The authors should indicate the number of samples measured, provide statistical data processing, and add error bars to the figures (especially Fig. 5). Without this, it is difficult to talk about a fundamental difference in the values. Only after determining the errors and justifying the data calculations can talk about the difference between the crystalline and amorphous phases.
  7. Figure 4(a) shows that the slopes of the straight lines for 50 and 60°C are virtually identical (especially if the straight line through the white dots is drawn correctly for the uppermost points), and determining the slope of these diffusion coefficient curves using the tangent will yield approximately the same values. However, in Figure 5, the authors' difference in these values, even on a stretched scale and without an error bar, appears to be maximal. Furthermore, Figure 4(b) does not contain data for temperatures of 40 and 50°C, and based on Figure 3, one would expect the difference for 50 and 60°C to be as minimal as in Figure 4(a). The authors should provide all the data and justify their calculations.
  8. Section 3.3 and Figure 6 – it is not entirely clear what the authors mean by maximum absorbance. How is this fundamentally different from the intensity of the absorption peak?

Round 2

Reviewer 4 Report

Comments and Suggestions for Authors

The authors answered most of the questions and made changes to the manuscript.

Author Response

We revised our manuscript in accordance with the reviewer’s comments. We believe these changes have significantly improved the quality of our work.